# The TB vaccine clinical trial centre directory: An inventory of clinical trial centres in Sub-Saharan Africa

**Puck T. Pelzer** [1,2,3]*, **Marit Holleman**[4], **Michelle E. H. Helinski**[5], **Ana Lucia Weinberg**[5†],
**Joeri Buis**[1], **Pauline Beattie**[5], **Thomas Nyirenda**[6], **Job van Rest**[1], **Gerald Voss**[4]

**1** KNCV Tuberculosis Foundation, The Hague, The Netherlands, **2** Amsterdam Institute for Global Health and Development (AIGHD), Amsterdam University Medical Centre, Amsterdam, The Netherlands, **3** London School of Hygiene and Tropical Medicine (LSHTM), London, United Kingdom, **4** TuBerculosis Vaccine Initiative (TBVI), Lelystad, The Netherlands, **5** European & Developing Countries Clinical Trials Partnership (EDCTP), The Hague, The Netherlands, **6** European & Developing Countries Clinical Trials Partnership (EDCTP), Cape Town, South Africa

† Deceased.
* ppelzer@iavi.org

**Data Availability Statement:** "All relevant data are within the paper and its Supporting Information files. The dataset used for this study is publicly available from the European & Developing Countries Clinical Trials Partnership (EDCTP)

## Abstract

### Background

There are over ten vaccine candidates for tuberculosis (TB) in the clinical pipeline that require testing in TB-prevalent populations. To accelerate the clinical development of TB vaccines, a directory of clinical trial centres was established in sub-Saharan Africa (SSA) to assess capacity for conducting late-stage TB vaccine trials.

### Methods

TB vaccine-related parameters were identified, and trial centres in SSA were identified and prioritized based on whether they had experience with TB or non-TB vaccine trials. A survey was sent to identified centres, and the resulting directory presents their capacity for TB vaccine trials. Centres that identified as eligible for TB vaccine trials also had the opportunity to participate to the survey. This article provides an overview of the TB vaccine clinical trial centre directory, including the number and distribution of centres, their general characteristics, and their experience with prior TB vaccine trials. It includes information on the capacity of the centres, such as laboratory biosafety level, patient support, and community engagement. It also includes a case study to demonstrate how the directory can be used to identify trial centres with specific capabilities needed for a particular TB vaccine trial.

### Results

Of the 134 identified centres, 56 provided information. Of these centres, 51 (91%) had phase 3 clinical trial experience and previous TB trials were conducted at 38 centres. Regarding TB vaccine trials, 19 centres conducted prevention of disease trials, 14 conducted prevention of infection trials, and 27 had no experience with TB vaccine clinical trials. From the respondents, 29 centers in South Africa were identified that could potentially

website: https://www.edctp.org/our-work/
coordination-tb-vaccine-funded-research/directory-
tb-vaccine-clinical-trial-sites-sub-saharan-africa/."

**Funding:** Development of tools and documents to
support the coordination of EDCTP TB vaccine
funded research'. The Funder had no role in study
design, data collection and analysis of the data. The
Funder was involved in the writing – review and
editing – of the manuscript.

**Competing interests:** The authors have declared
that no competing interests exist.

conduct TB vaccine trials, followed by Tanzania (5), Kenya (5), Nigeria (3), and Uganda and Ethiopia (2 each). Trial sites in other countries were underrepresented, based on this survey.

## Conclusion

The establishment of a clinical trial centre directory can provide a basis for decision-making by various stakeholders. Despite some limitations in survey methodology, the findings suggest opportunities for expanding the evaluation of clinical trial capacity in other disease-prevalent countries and continents. Such data would be valuable in further enriching the Clinical Trial Community which a resource that geographically highlights clinical trial investments and capacities in African research ecosystem.

## Summary points

- New TB vaccine candidates need to be assessed in clinical trials in countries with high rates of TB in the coming years.

- An open-access directory of TB vaccine clinical trial centres in sub-Saharan Africa was established, providing an overview of the capacity to conduct clinical trials for TB vaccine candidates (http://www.edctp.org/our-work/coordination-tb-vaccine-funded-research/directory-tb-vaccine-clinical-trial-sites-sub-saharan-africa/).

- The directory is intended for clinical triallists, funders, policymakers, and researchers to accelerate the clinical development of novel TB vaccines by providing useful information.

- Regular updates are necessary to ensure the directory remains relevant for vaccine development and feeds into the continental Clinical Trials Community (https://ctc.africa/).

## Introduction

Tuberculosis (TB) remains a significant global health threat, responsible for approximately 1.3 million deaths in 2020 alone [1], with the African region accounting for around a quarter of new TB cases worldwide. The burden of TB is especially acute in Sub-Saharan Africa (SSA). In this region, 16 of the 30 countries with the highest TB burden globally are located here [2, 3]. Despite a global reduction in TB incidence, SSA has seen a slower decline compared to the global average, highlighting an urgent need for targeted interventions [4].

Immunization against TB is a critical component of the fight against this disease, yet the only registered vaccine currently available is the bacillus Calmette Guerin vaccine (BCG) [5]. Although the BCG vaccine has been shown to prevent certain types of TB (i.e., meningitis and disseminated TB in young children), its protection against pulmonary TB is variable [6, 7]. Currently BCG is only recommended for new-borns and is contra-indicated in HIV-positive populations. Given the ongoing threat of TB and the ambitious "End TB" goals set out by the World Health Organization, the development of new and improved vaccines is urgently needed to accelerate progress towards elimination.

In SSA, the intersection of TB and HIV presents a major public health challenge, as the region has one of the highest rates of HIV-TB co-infection worldwide [3]. HIV-positive

individuals are at significantly higher risk of developing active TB, and their immune responses to vaccines can differ from those of HIV-negative individuals [3]. Furthermore, SSA is characterized by genetic diversity, both in human populations and in the strains of *Mycobacterium tuberculosis* circulating in the region. This diversity makes it essential to test new TB vaccines in SSA to ensure their effectiveness across different genetic backgrounds and pathogen strains.

The current pipeline includes promising vaccine candidates at various stages of clinical trails, ranging from recombinant viral vectors, live attenuated *Mycobacterium tuberculosis*, improved BCGs, inactivated whole cells, and adjuvanted proteins [8]. At present, five vaccine candidates are in phase 3 clinical trials, with two additional candidates soon entering phase 3. VPM1002 for example, targets prevention of infection (PoI), prevention of disease (PoD) and prevention of recurrence (PoR) in different phase 3 trials. Other candidates, such as MIP/ Immuvac, GamTBvac and MTBVAC are focused on PoD. The M72/AS01E candidate is scheduled for a phase 3 efficacy trial for PoD and PoI. BCG revaccination will be evaluated in a Phase 3 trial for PoI [9].

Despite these advancements, the capacity for conducting TB vaccine trials remains limited globally, particularly in regions like SSA, which are most affected by the disease [10]. Conducting TB vaccine clinical trials in SSA is important for several reasons. First, the high TB burden in the region provides the opportunity to test the efficacy of new TB vaccines in real-world settings where the disease is most prevalent. Second, SSA also has a high prevalence of HIV, which is a major risk factor for developing active TB. The high prevalence of HIV in SSA necessitates vaccine trials that can assess the performance of TB vaccines in this high-risk population. Third, the region's genetic and pathogen diversity requires vaccine efficacy to be evaluated across a wide range of backgrounds to ensure global applicability. Lastly, SSA's healthcare infrastructure, which often faces resource constraints, provides critical insights into the operational feasibility of vaccine deployment under real-world conditions.

Conducting TB vaccine trials in SSA promotes equity in global health research by ensuring that the populations most affected by the disease are not left behind in the development and rollout of new interventions. Mapping the clinical vaccine trial sites with capacity to conduct TB vaccine trials provides valuable insights in the current status of SSA to conduct trials, promotes site selection, and is essential for advocating for the expansion of trial capacity in SSA.

The capacity needed for TB vaccine trials differs significantly from the requirements of trials for TB diagnostics, drugs, and preventive treatment. TB vaccine trials are conducted in healthy populations or infected populations, while trials for drugs and diagnostics require TB patients. Clinical trials for preventive treatment require populations with high TB infection rates or a substantial TB-HIV co-infection prevalence. For TB vaccine clinical trials, follow-up periods typically range from several months to several years, depending on the specific vaccine candidate and study design. For TB diagnostic tests, follow-up periods in clinical trials can vary widely based on the type of diagnostic being evaluated. They may range from a few weeks to several months. Furthermore, the number of participants needed for TB vaccine trials is substantially higher. To cover the host and bacteriological genetic variety, multiple locations are required for phase 3 TB vaccine trials to allow the licensing of new vaccines in different countries. In addition, trial centres require diverse and specific capacity and capabilities for the different targeted endpoints. For example, PoD trials require a substantially larger sample size with prolonged follow-up, as compared to PoI trials [10].

Lessons learned from previous TB vaccine efficacy trials highlight the importance of innovative trial designs, selecting appropriate efficacy endpoints, and carefully choosing sites based on epidemiological data and target population [10]. Performing such work requires access to clinical trial centres with adequate capacity, particularly for phase 3 licensure trials however,

currently such information is not widely available. To address this gap, further advance TB vaccine development, and promote site selection,. we curated a directory of clinical centres in SSA that are suitable for future TB vaccine studies, with possibility of regular updates.

## Methods

To identify trial centres and gather information on key parameters, SSA a multistep approach was employed by triangulating and complementing information from both online resources, networks, and social media.

For the identification of the SSA centres, we triangulated information from a desk review and online sources including ClinicalTrials.gov, Centerfinder, Global Health Trials, Centrewatch, AAS trial centres, and AGT network. We asked input from NTP managers, other experts, and made use of the EDCTP networks to identify additional centres and centres that potentially, did not have an online presence. Additionally in a second round of data collection, we used social media so that centres who were not yet identified but eligible for conducting TB vaccine trials could also be identified and reviewed for inclusion

For the identification of essential parameters for the TB vaccine trials, a desk review was conducted to review the existing literature on TB vaccine trials and reports. The resulting survey and directory parameters consisted of the centre characteristics, the type of disease targets and intervention, and expertise and the capacity (Table 1). A clinical trial centre directory template was developed along with data collection tools, including an Excel file and online survey. Data collection was carried out using a Microsoft Forms online survey tool, which contained 40 questions in English (S1 File). Four experts with knowledge on and/or experience with TB vaccine trials, reviewed the parameters, the data collection tool, and the survey design.

To identify clinical trial centres for outreach,. Each centre was evaluated for suitability for the TB vaccine trial centre directory, with shortlisting based on prior experience with TB vaccine trials or trials related to TB and non-TB vaccine trials. In addition, centres were included based on expert recommendation, and contact persons were identified through online searches and referrals from the field. Centres that considered themselves eligible for conducting TB vaccine trials were also included.

The respective contact persons for the shortlisted centres were invited by email to complete the online survey, with survey responses collected between April and August 2020. Centres that did not respond received up to two reminders, and additional contact details were sought to solicit a response. The first round of data collection closed in 2020, and in 2022, the survey was reopened to include clinical trial centres that considered themselves eligible or wished to be included in the directory.

The survey data were cleaned using Statacorp v15.1 [11]. Data from each centre were manually inspected, duplicates were removed, and data entry errors were corrected. The respondents were asked to validate the data if necessary. The clinical trial centre directory entries were summarized by general centre characteristics and TB vaccine trial-specific characteristics. For each, the potential trial endpoints, recruitable population, and phase of clinical trials were highlighted, along with TB incidence in the country as reported by WHO (8) and population and country size from World Bank estimates (9) for all countries in SSA and for included trial centres.

## Results

The survey data were compiled into the TB vaccine clinical trial centre directory. The data is publicly accessible on the EDCTP website. Fig 1 shows the website of the trial centre directory. Users can review information on clinical trial centres in each country by clicking on an

**Table 1. Parameters considered in the TB vaccine clinical trial centre directory: Centre characteristics, type of disease targets, intervention expertise and capacity, TB vaccine.**

| Clinical trial centre information | Location |
|---|---|
| | Type of health facilities |
| | Single or multiple clinical trial centre |
| | Standalone or multiple location |
| | Year of establishment |
| | Public or private |
| | Profit/ non-profit |
| | Government of non-government |
| | Principal funding source |
| | Supported by EDCTP |
| | Working language |
| | Completed clinical trials |
| **Type of diseases / interventions** | Therapeutic area |
| | Endpoint of prior trials conducted (vaccines, diagnostics, treatment regimens) |
| **Expertise and capacity** | Trial phase capacity |
| | Recruitable participant population |
| | Available clinical trial centre population data |
| | Research capacity |
| | Clinical testing facilities |
| | Clinical testing capacity |
| | Laboratory capacity |
| | Community engagement |
| | Patient support |
| | Side effect management |
| | Follow up capacity |
| | Certification |
| **TB vaccine specific** | Endpoint of TB trials conducted (vaccines, diagnostics, treatment regimens) |
| | TB vaccine target |
| | TB vaccine target population |

interactive map to review key details for that location. In addition, the entire dataset including additional parameters and the data dictionary are available as downloads from the website.

## General characteristics

Our study shortlisted 134 clinical trial centres in SSA, of which 43% responded to our survey (S1 Table) and were included in the trial centre directory. The first version of the directory, which included data from 45 centres, was established in May 2021. Between May 2021 and June 2022, an additional 11 centres were included based on self-assessment. The directory was last updated in June 2022.

The majority of the 56 centres in the directory are located in South Africa (29), followed by Tanzania and Kenya with five centres each, and three in Nigeria. Two centres are in Uganda and Ethiopia, and the remaining nine countries have one centre each (Table 2 and see S2 Table for the centres' cities). Out of the 56 centres, 22 were standalone clinical trial centres and 33 were clinical trial centres with multiple sites. The year of establishment of these centres ranges between 1934 and 2021. Most of the centres (46) were non-profit organisations. On average, the assessed centres had completed 26 clinical trials (TB or non-TB) and had follow-

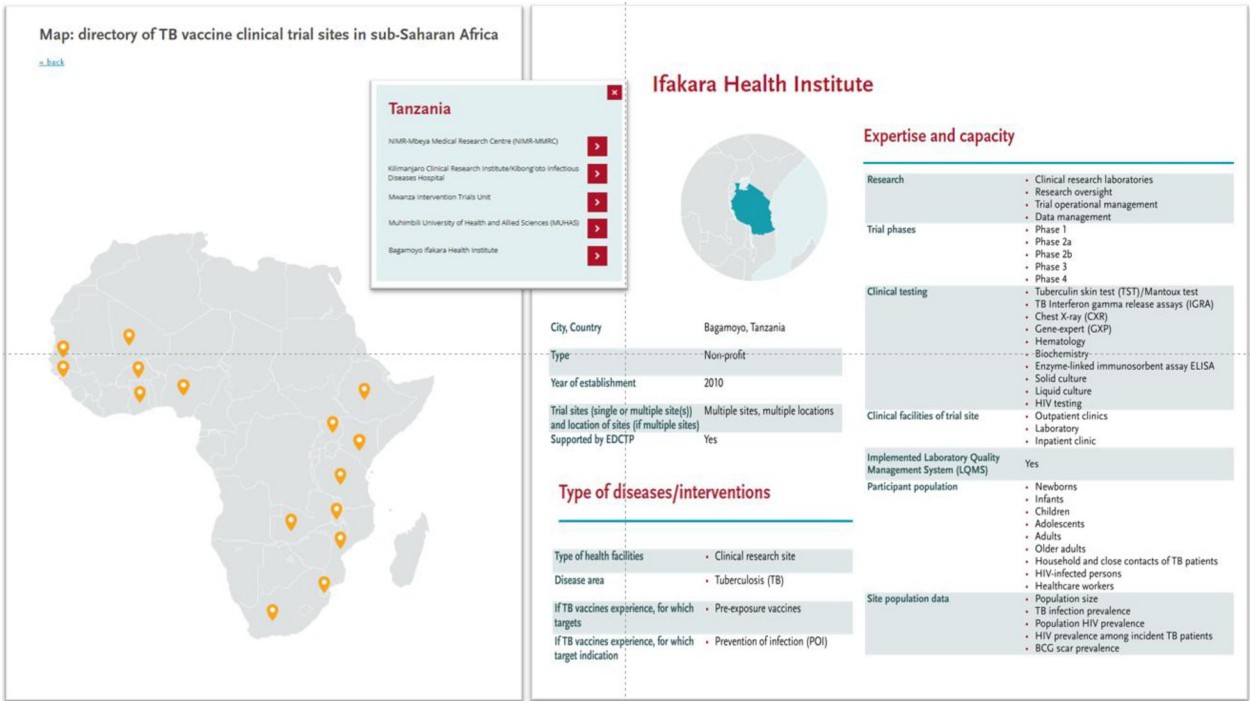

**Fig 1. Website of the TB vaccine trial centre directory, and an example of key trial centre information provided online.**

up capacity of 10 years or more at 20 centres, 5 years at 21 centres, and less than 3 years at 13 centres (Fig 2, Panel A). Patient support, community engagement, and side effect management were reported to be in place at most of the centres (Fig 2, Panel B). About one-third of the centres (24) reported having laboratory biosafety level (BSL) 3 capabilities, and four had BSL4 capabilities at the centre or affiliated laboratory (Fig 2, Panel C). Most centres were clinical research centres (51), while 19 were academic institutions (Fig 2, Panel D). Thirty-eight centres had a clinical research focus on TB (Fig 2, Panel E).

Twenty-seven clinical trial centres reported prior TB vaccine l trial experience, with the majority of these having experience in TB vaccine trials in adult populations. (Fig 3, Panel A). Twenty centres had capacity for preexposure TB vaccines and 17 for post (Fig 3, Panel B). Most of the 27 centres had experience with PoD (19) and/or PoI (14) trial endpoint capacity (Fig 3, Panel E). Tuberculin skin testing was available at 46 centres, interferon gamma release assay at 32 centres, culture testing at 30 centres and HIV testing at 53 centres (Fig 3, Panel D).

From the respondents, South Africa had the most centres with experience in TB vaccine trials (16 centres), followed by Tanzania (3 centres), Kenya (2 centres) and Ethiopia, Guinea-Bissau, Mozambique, Senegal, Uganda and Zambia with one centre each (Table 2). PoD trial endpoint capacity was available in 19 centres, of which the majority was in South Africa. The same was observed for PoI trial endpoint capacity, with 10 out of 14 centres being located in South Africa. PoR capacity was mostly also seen in South Africa (7 out of 9 centres). Of all the trial centres, 51 had phase 3 capacity (any disease). In terms of recruitable populations, centres had experience in working with adult populations. Less experience was among adolescents (46/56) and least experience with neonates (26/56).

Fig 4 shows an overview of WHO TB incidence rates per 100,000 in SSA, highlighting potential TB vaccine clinical trial centres included in the directory. The six countries with the highest relative incidence rate per 100,000 in SSA either had no trial centre identified

**Table 2. Clinical trial centre capacity per country.**

| Country | Trial centres included | Identified | Responded | Experience with TB vaccine trials | | | Experience with trial phases (any disease/ intervention) | | | | | Recruitable population | | | |
|---|---|---|---|---|---|---|---|---|---|---|---|---|---|---|---|
| | | | | PoD | PoI | PoR | I | II a | II b | III | IV | Adults | Adolescents | Children | Neonates |
| Angola | 0 | 0 | 0 | | | | | | | | | | | | |
| Benin | 0 | 1 | 0 | | | | | | | | | | | | |
| Botswana | 0 | 2 | 0 | | | | | | | | | | | | |
| Burkina Faso | 1 | 1 | 1 | 0 | 0 | 0 | 0 | 1 | 1 | 1 | 1 | 1 | 1 | 1 | 1 |
| Burundi | 0 | 0 | 0 | | | | | | | | | | | | |
| Cameroon | 1 | 3 | 1 | 0 | 0 | 0 | 0 | 0 | 1 | 1 | 1 | 1 | 1 | 0 | 0 |
| Cape Verde | 0 | 0 | 0 | | | | | | | | | | | | |
| Central African Repulic | 0 | 0 | 0 | | | | | | | | | | | | |
| Chad | 0 | 0 | 0 | | | | | | | | | | | | |
| Comoros | 0 | 0 | 0 | | | | | | | | | | | | |
| Democratic Republic of the Congo | 0 | 1 | 0 | | | | | | | | | | | | |
| Djibouti | 0 | 0 | 0 | | | | | | | | | | | | |
| Equitoral Guinea | 0 | 0 | 0 | | | | | | | | | | | | |
| Eritrea | 0 | 0 | 0 | | | | | | | | | | | | |
| Eswatini | 1 | 1 | 1 | 0 | 0 | 0 | 0 | 1 | 1 | 0 | 0 | 1 | 1 | 1 | 1 |
| Ethiopia | 2 | 3 | 2 | 1 | 0 | 0 | 2 | 2 | 2 | 2 | 1 | 2 | 2 | 0 | 0 |
| Gabon | 0 | 1 | 0 | | | | | | | | | | | | |
| Gambia | 0 | 2 | 0 | | | | | | | | | | | | |
| Ghana | 1 | 3 | 1 | 0 | 0 | 0 | 0 | 1 | 1 | 1 | 1 | 1 | 1 | 1 | 1 |
| Guinea | 0 | 0 | 0 | | | | | | | | | | | | |
| Guinea-Bissau | 1 | 1 | 1 | 0 | 0 | 0 | 0 | 0 | 1 | 1 | 1 | 1 | 1 | 1 | 1 |
| Ivory Coast | 0 | 0 | 0 | | | | | | | | | | | | |
| Kenya | 5 | 13 | 5 | 1 | 2 | 1 | 4 | 4 | 3 | 4 | 4 | 5 | 5 | 3 | 3 |
| Lesotho | 0 | 0 | 0 | | | | | | | | | | | | |
| Liberia | 0 | 0 | 0 | | | | | | | | | | | | |
| Madagascar | 0 | 1 | 0 | | | | | | | | | | | | |
| Malawi | 1 | 5 | 1 | 0 | 0 | 0 | 0 | 0 | 0 | 1 | 1 | 1 | 1 | 1 | 1 |
| Mali | 1 | 1 | 1 | 0 | 0 | 0 | 1 | 1 | 0 | 0 | 0 | 1 | 1 | 1 | 0 |
| Mauritania | 0 | 0 | 0 | | | | | | | | | | | | |
| Mozambique | 1 | 1 | 1 | 1 | 0 | 0 | 0 | 1 | 1 | 1 | 1 | 1 | 1 | 1 | 1 |
| Namibia | 0 | 0 | 0 | | | | | | | | | | | | |
| Niger | 0 | 0 | -0 | | | | | | | | | | | | |
| Nigeria | 3 | 10 | 3 | 0 | 0 | 0 | 1 | 1 | 2 | 3 | 2 | 3 | 2 | 1 | 1 |
| Republic of the Congo | 0 | 0 | 0 | | | | | | | | | | | | |
| Rwanda | 0 | 2 | 0 | | | | | | | | | | | | |
| Sao Tomé and Príncipe | 0 | 0 | 0 | | | | | | | | | | | | |
| Senegal | 1 | 2 | 2 | 0 | 0 | 0 | 1 | 1 | 1 | 1 | 0 | 1 | 1 | 1 | 1 |
| Seychelles | 0 | 0 | 0 | | | | | | | | | | | | |
| Sierra Leone | 0 | 0 | 0 | | | | | | | | | | | | |
| South Africa | 29 | 50 | 29 | 14 | 10 | 7 | 13 | 21 | 27 | 27 | 16 | 29 | 21 | 13 | 10 |
| South Sudan | 0 | 0 | 0 | | | | | | | | | | | | |
| Tanzania | 5 | 8 | 5 | 1 | 2 | 1 | 3 | 3 | 5 | 5 | 4 | 5 | 4 | 4 | 4 |
| Togo | 0 | 0 | 0 | | | | | | | | | | | | |

*(Continued)*

**Table 2.** (Continued)

| Country | Trial centres included | Identified | Responded | Experience with TB vaccine trials | | | Experience with trial phases (any disease/ intervention) | | | | | Recruitable population | | | |
|---|---|---|---|---|---|---|---|---|---|---|---|---|---|---|---|
| Uganda | 2 | 13 | 2 | 0 | 0 | 0 | 1 | 1 | 1 | 2 | 1 | 2 | 2 | 1 | 1 |
| Zambia | 1 | 6 | 1 | 1 | 0 | 0 | 0 | 0 | 1 | 1 | 1 | 1 | 1 | 0 | 0 |
| Zimbabwe | 0 | 3 | 0 | | | | | | | | | | | | |
| Total | 56 | 134 | 57 | 19 | 14 | 9 | 26 | 38 | 48 | 51 | 35 | 56 | 46 | 30 | 26 |

PoD = prevention of disease, PoI = prevention of infection, PoR = prevention of recurrence

(Republic of Congo, Lesotho, Angola, Central African Republic, Namibia), or had one centre identified that did not respond (Gabon). Of the included trial centres, South Africa has the highest incidence relative to the population and also the largest number of clinical trial centres. The four subsequent countries with the highest incidence were Mozambique, Guinea-Bissau, Zambia and Eswatini (S1 Table). Each of these countries has one trial centre that responded to the survey, potentially suitable for TB vaccine trials in the country. It should be noted here that Eswatini and Guinea-Bissau are relatively small in terms of population size, and for Zambia five additional centres were identified, but these did not respond to the survey.

One of the applications of the directory is to identify trial centres that have a specific combination of capabilities needed for a particular TB vaccine trial. Fig 5 shows the applicability of the trial centre directory for a case study that would require phase 3 trial capabilities, with experience in the PoD trial endpoint among adolescent in a WHO high TB burden country, where infection testing would be carried out through IGRA and GeneXpert for TB. Participants would be approached through an in-patient clinic. Applying these search criteria in the clinical trial centre directory resulted in five potential centres that could be used, as shown in Fig 5.

## Discussion

This project curated a directory of clinical trial centres capable of conducting TB vaccine trials in SSA. The evaluation of the compiled trial centre directory highlights the need to develop more trial centres across the four regions of SSA reducing reliance on the South African sites. This is especially relevant considering the pipeline of promising vaccine candidates that are being prepared for phase 3 studies [12]. In terms of target populations, the evaluated centres had sufficient capacity to recruit adults, but the limitations on neonates was most striking. Therefore, it is necessary to strengthen the capacity of the trial centres to conduct TB vaccine trials for different indications.

The clinical trial centre assessment had some limitations. In the directory, centers without formal documentation and/or low resources are potentially underrepresented in this directory, of which the extent is unknown. However, we have tried to minimize the extent by complementing online sources with knowledge from internal networks and local NTPs. Only responding sites were included, while in some countries additional trial centres were identified, making it difficult to have a comprehensive overview for any such given country. There is over-representation of South African centres, which could be attributed to the country's investments in TB vaccine centres. TB vaccine trials need to be conduced in areas with high TB incidence, which are typically in low-resource settings that lack capacity and prior trial experience. The marked prevalence of prior studies in South Africa positions it as an outlier in terms of capacity, elucidating the distinctive nature of its contribution to TB vaccine research. This concentration of research activity in South Africa emphasizes the needs for increased

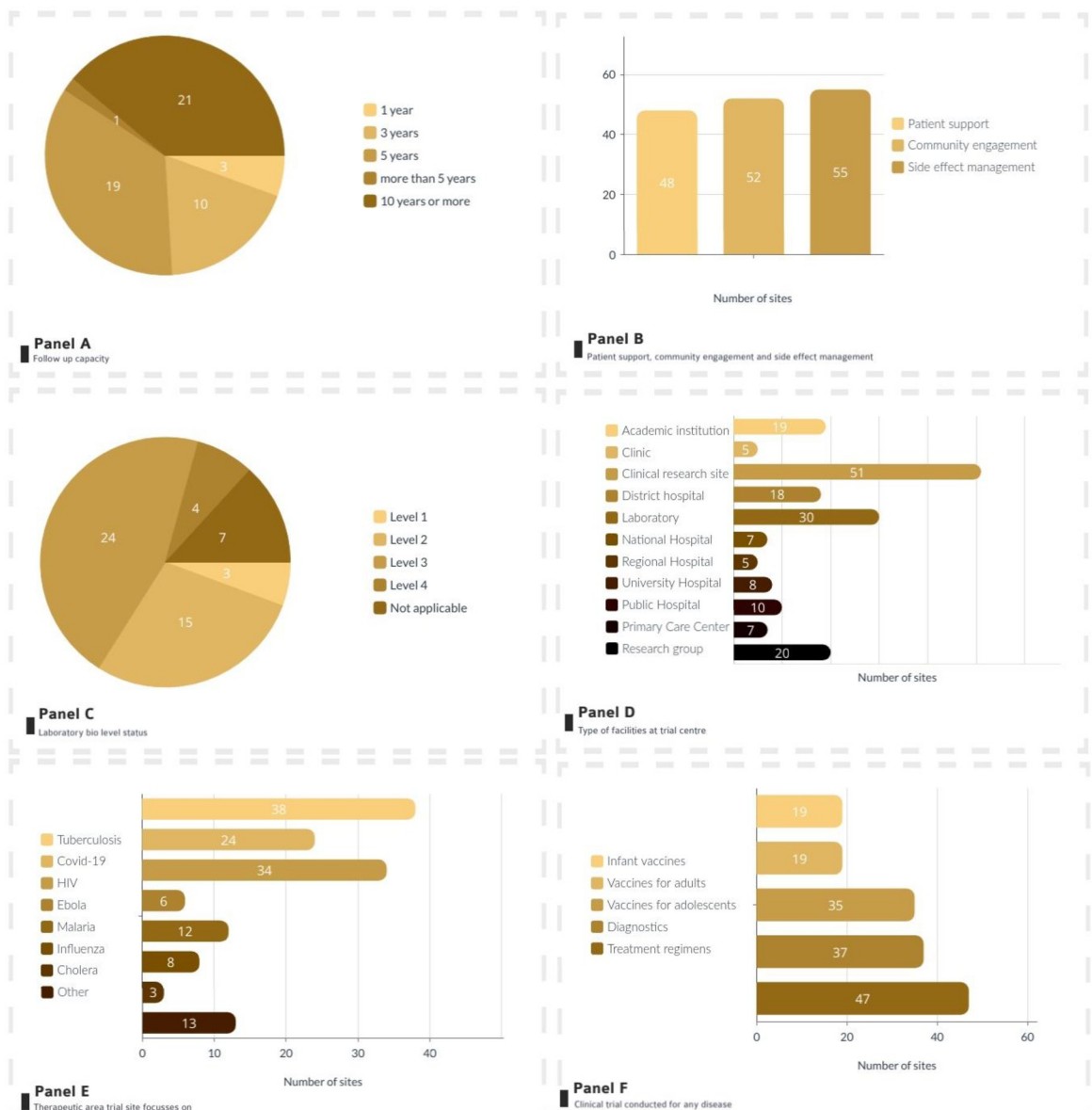

**Fig 2. Overview of centre characteristics overview of trial centres included in the directory.** Panel A: The follow-up capacity of the trial centre | Panel B: patient support, community engagement and side effect management | Panel C Laboratory biosafety level | Panel D: Type of facilities at trial centre| Panel E: Therapeutic area trial centre focusses on | Panel F: Clinical trials conducted for any disease. Panel B,D,E, F are not mutually exclusive.

capacity needs for centres across Africa. In addition, the information provided was self-assessment by the centres and not further validated. Just under half of the centres who were contacted, responded (43%). Possible reasons affecting the response rate could have been language barriers, timing of survey during the COVID-19 pandemic when research centres were prioritising response to COVID, difficulties identifying the most appropriate contact person for the centre and staff turnover. Future efforts in this area should aim to include centres and countries that were missed as well as an expansion to other geographical (global) areas including translating the survey in other languages e.g., French and Portuguese. This will be a necessary

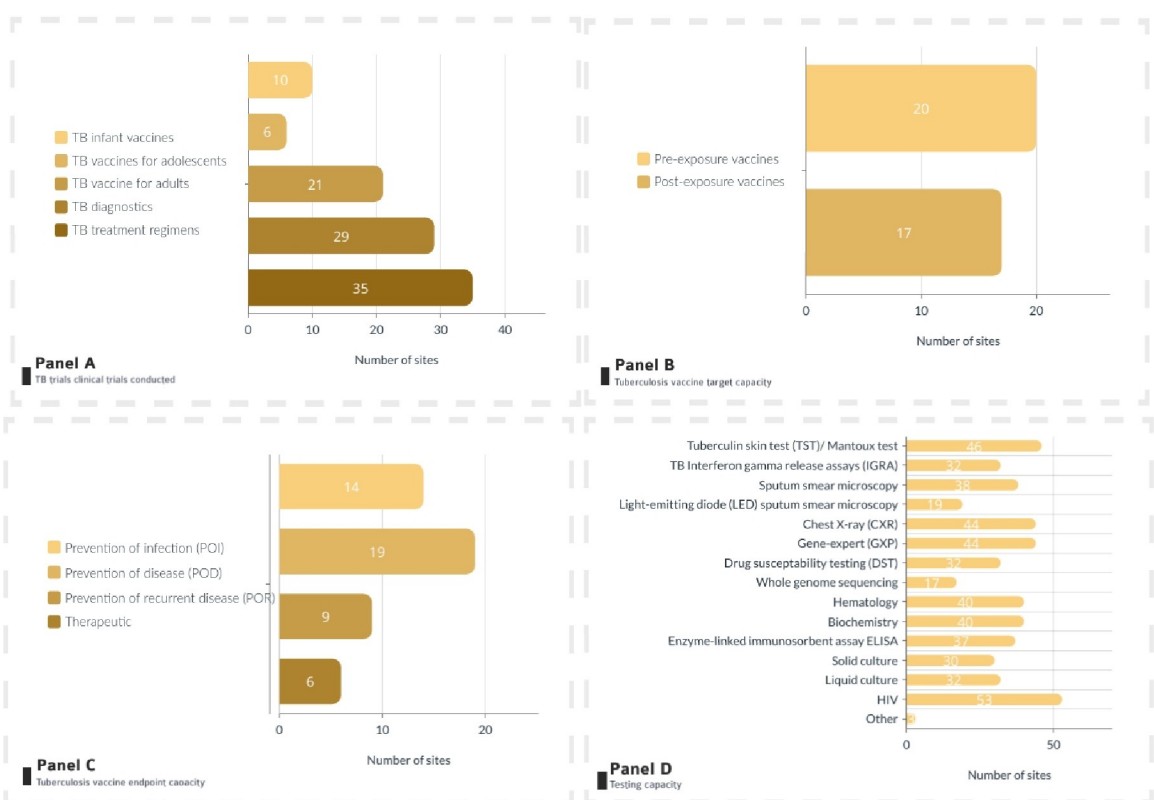

**Fig 3. Overview of TB clinical trial specific characteristics overview of trial centres included in the directory.** Panel A: TB trials clinical trials conducted | Panel B: Tuberculosis vaccines target capacity | Panel C: Tuberculosis vaccines endpoint | Panel D: Testing capacity. Panel A-D are not mutually exclusive.

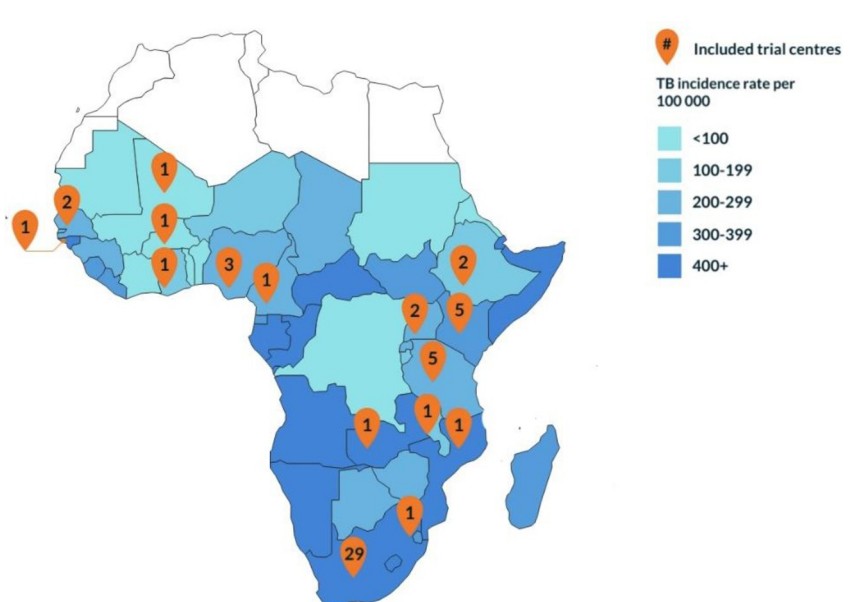

**Fig 4. Map of TB vaccine trial centres included in the directory and WHO TB incidence rates per 100 000 population.**

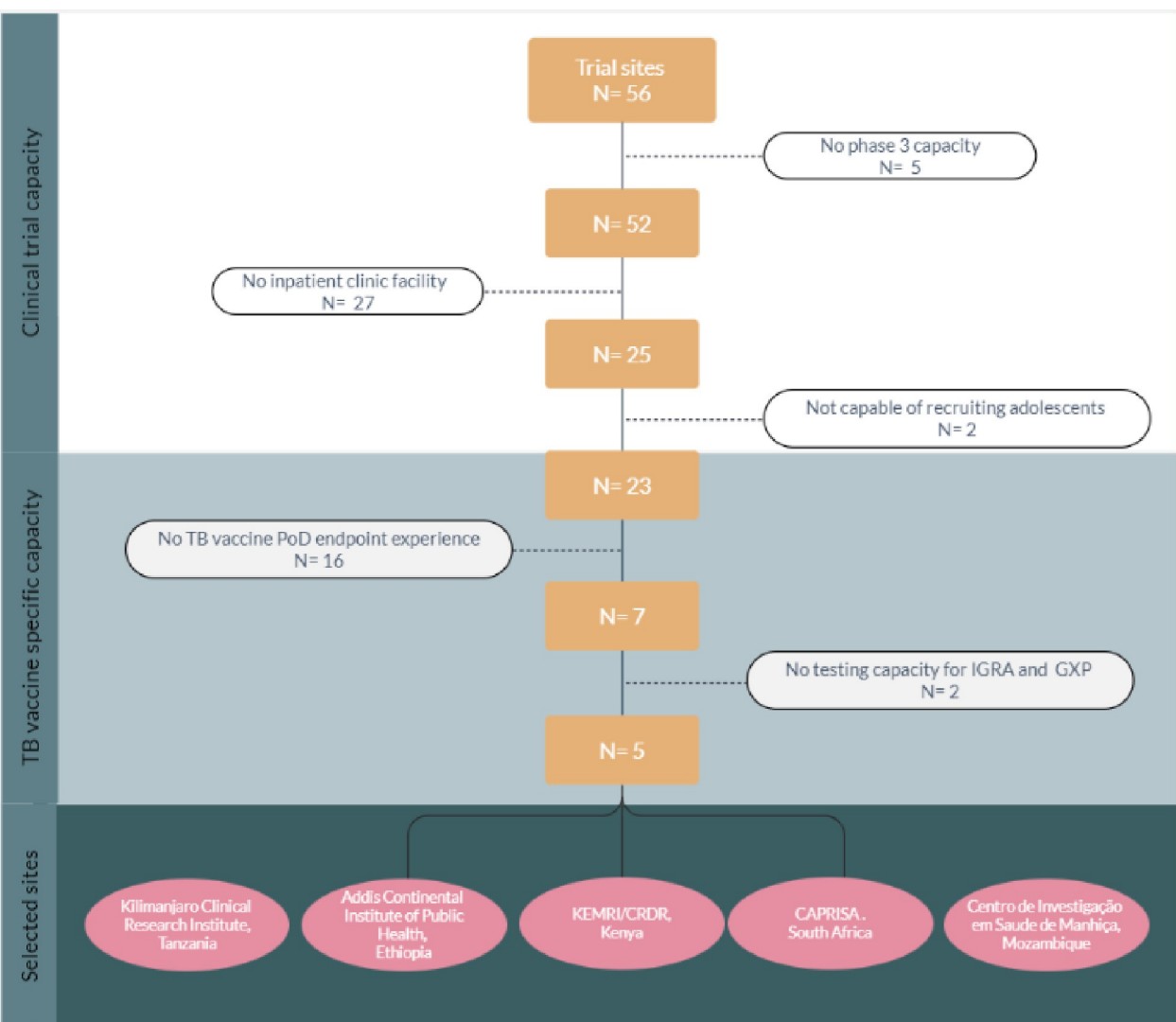

**Fig 5. Case study using the TB vaccine clinical trial centre directory.**

effort, as TB vaccine trials will be conducted in countries outside of Africa where TB is endemic.

To keep the TB vaccine clinical trial centre directory relevant and useful, regular updates are necessary. In terms of epidemiological indicators, this mapping assessment was limited to national data. Ideally, centres should be evaluated by their own age-specific epidemiological indicators [12]. In terms of centre capacity, including diagnostics, this may evolve over time and to remain relevant, centres should be regularly contacted to update information.

Despite the limitations described above the directory allows to identify the registered centres that have experience and capacity in terms of TB vaccines, complementing other initiatives to map clinical trial centers such as the African clinical trials community (https://www.ctc.africa/about), which is a database of clinical trials investments and clinical sites including their capacities for all diseases. The study did not investigate the factors that influence the establishment of TB vaccine trial centres in individual countries. Thus, further work should also investigate what incentivises institutions to become a TB vaccine trial centre. For example, prior

evaluations of TB vaccine trial centres indicated that difficulties with logistics and operations could pose a problem in terms of trial execution [13, 14]. Kaguthi et al. [15] discuss lessons learned from development of a TB vaccine trial centre for design and implementation of trials in Africa. They highlight that performing exhaustive epidemiological research to offer background incidence and prevalence rates of TB for clinical trials, and the inclusion of investigators in discussions about tests to be run on bio-banked samples are some of the key lessons learned in centre development.

Our study focused on SSA to demonstrate efficacy under varying complex epidemiological and clinical conditions, but for the global development of TB vaccines, other countries and populations must be included in late-stage TB vaccine trials. In a recent study [16], experts indicated that nationally conducted trials could be a requirement in the country prior to implementation, indicating the importance of centre development throughout the globe and especially in TB endemic areas in order to ensure adequate human and mycobacterial diversity. Therefore, the expansion of the clinical trial centre directory for TB vaccines to other geographies may prove useful.

As the first directory of its kind, this comprehensive listing of clinical trial centres in Sub-Saharan Africa (SSA) offers critical insights that can significantly impact decision-making on resource allocation and site selection for TB vaccine trials. Policymakers, research funders, and global health organizations can utilize this directory to prioritize sites with the necessary capacity and relevant experience. Furthermore, the directory facilitates improved coordination of research efforts across the region by identifying gaps in capacity and promoting collaboration between trial centres. This can help streamline the selection of trial sites, minimize duplication of efforts, and ensure that research is conducted efficiently across multiple locations. Additionally, the directory highlights countries that may require further investment to meet the demands of large-scale phase 3 trials, thus strengthening the research infrastructure in SSA. Ultimately, this directory serves as a vital tool for building a more coordinated and robust TB vaccine trial network in SSA, accelerating progress towards the global goal of ending TB.

## Conclusions

A survey of centres in TB endemic countries through an online questionnaire resulted in the establishment of a TB vaccine clinical trial centre directory providing information that may be helpful to accelerate the clinical development of TB vaccines. The directory is intended as a tool for clinical trialists, vaccine developers, funders, policymakers, and researchers. It provides a snapshot in time and needs to be updated periodically to remain relevant. The capacity gaps exposed by the directory showed that there is a need to continue strengthening existing TB vaccine trial centres as well as establishing multiple settings, especially where TB prevalence is high. A next useful step will be the creation of a global directory of TB vaccine trial centres to reflect the need for TB vaccine development in geographies other than SSA.

## Supporting information

**S1 File. TB vaccine clinical trial site directory.**
(PDF)

**S1 Table. Overview of centres contacted, and centres responded per country by TB incidence, population size, and country size.** WHO TB incidence rates [17] | Worldbank population [18].
(DOCX)

**S2 Table. Overview of the cities in which the centres were identified.**
(DOCX)

## Acknowledgments

The directory of TB vaccine clinical trial centres suitable for future TB vaccine studies in Sub-Saharan Africa was conducted by TBVI and KNCV Tuberculosis foundation. The following experts provided advice on essential parameters that were included in the survey: Suzanne Verver, Frank Cobelens, Richard White and Rebecca Harris. We thank Ieva Leimane for her help with survey design, Daniela Pereira for data visualization, Degu Dare, Jeremy Hill, and Max Meis for technical support, and Stephanie Borsboom for project management. Our assessment relied partly on work done by KNCV for Aeras. Last but not least, we wish to thank all contact points from the centres for providing information

## Author Contributions

**Conceptualization:** Puck T. Pelzer, Marit Holleman, Michelle E. H. Helinski, Ana Lucia Weinberg.

**Data curation:** Puck T. Pelzer.

**Formal analysis:** Puck T. Pelzer.

**Funding acquisition:** Marit Holleman, Gerald Voss.

**Methodology:** Puck T. Pelzer, Gerald Voss.

**Project administration:** Puck T. Pelzer, Marit Holleman, Ana Lucia Weinberg.

**Resources:** Michelle E. H. Helinski.

**Software:** Puck T. Pelzer.

**Supervision:** Puck T. Pelzer, Gerald Voss.

**Validation:** Puck T. Pelzer.

**Visualization:** Puck T. Pelzer, Job van Rest.

**Writing – original draft:** Puck T. Pelzer.

**Writing – review & editing:** Marit Holleman, Michelle E. H. Helinski, Ana Lucia Weinberg, Joeri Buis, Pauline Beattie, Thomas Nyirenda, Job van Rest, Gerald Voss.

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
