## [Decision Letter · Decision Letter 0]

1 Dec 2023

PONE-D-23-31310The TB vaccine clinical trial centre directory: an inventory of clinical trial centres in sub-Saharan AfricaPLOS ONE

Dear Dr. Pelzer,

Thank you for submitting your manuscript to PLOS ONE. After careful consideration, we feel that it has merit but does not fully meet PLOS ONE’s publication criteria as it currently stands. Therefore, we invite you to submit a revised version of the manuscript that addresses the points raised during the review process.

Please note that we have only been able to secure a single reviewer to assess your manuscript. We are issuing a decision on your manuscript at this point to prevent further delays in the evaluation of your manuscript. Please be aware that the editor who handles your revised manuscript might find it necessary to invite additional reviewers to assess this work once the revised manuscript is submitted. However, we will aim to proceed on the basis of this single review if possible.

We look forward to receiving your revised manuscript.

Kind regards,

Jianhong Zhou

Staff Editor

PLOS ONE

“Development of tools and documents to support the coordination of EDCTP TB vaccine funded research.”

“The directory of TB vaccine clinical trial centres suitable for future TB vaccine studies in Sub-Saharan Africa was commissioned and funded by the EDCTP2 programme supported by the European Union and conducted by TBVI and KNCV Tuberculosis foundation. The following experts provided advice on essential parameters that were included in the survey: Suzanne Verver, Frank Cobelens, Richard White and Rebecca Harris. We thank Ieva Leimane for her help with survey design, Daniela Pereira for data visualization, Degu Dare, Jeremy Hill, and Max Meis for technical support, and Stephanie Borsboom for project management. Our assessment relied partly on work done by KNCV for Aeras. Last but not least, we wish to thank all contact points from the centres for providing information.”

“Development of tools and documents to support the coordination of EDCTP TB vaccine funded research.”

4. We noted in your submission details that a portion of your manuscript may have been presented or published elsewhere. [The background data is publicly available on the EDCTP website

https://www.edctp.org/our-work/coordination-tb-vaccine-funded-research/directory-tb-vaccine-clinical-trial-sites-sub-saharan-africa/] Please clarify whether this publication was peer-reviewed and formally published. If this work was previously peer-reviewed and published, in the cover letter please provide the reason that this work does not constitute dual publication and should be included in the current manuscript.

6. We note that Figures 1 and 4 in your submission contain [map/satellite] images which may be copyrighted. All PLOS content is published under the Creative Commons Attribution License (CC BY 4.0), which means that the manuscript, images, and Supporting Information files will be freely available online, and any third party is permitted to access, download, copy, distribute, and use these materials in any way, even commercially, with proper attribution. For these reasons, we cannot publish previously copyrighted maps or satellite images created using proprietary data, such as Google software (Google Maps, Street View, and Earth). For more information, see our copyright guidelines: http://journals.plos.org/plosone/s/licenses-and-copyright.

1. You may seek permission from the original copyright holder of Figures 1 and 4 to publish the content specifically under the CC BY 4.0 license. 

7. We notice that your supplementary 2 table are included in the manuscript file. Please remove them and upload them with the file type 'Supporting Information'. Please ensure that each Supporting Information file has a legend listed in the manuscript after the references list.

Reviewers' comments:

Reviewer's Responses to Questions

**Comments to the Author**

1. Is the manuscript technically sound, and do the data support the conclusions?

Reviewer #1: Yes

2. Has the statistical analysis been performed appropriately and rigorously? 

Reviewer #1: Yes

3. Have the authors made all data underlying the findings in their manuscript fully available?

Reviewer #1: Yes

4. Is the manuscript presented in an intelligible fashion and written in standard English?

Reviewer #1: Yes

5. Review Comments to the Author

Reviewer #1: This is an important manuscript establishing a directory of clinical trial centers across sub-Saharan Africa to assess capacity for conducting late-stage TB vaccine trials. Considering this is the primary location of most future clinical trials this manuscript has important relevance and I recommend publishing. What I find important is the new information provided (available centers) but, perhaps just as valuable, also is locations where centers are not available and capacity is needed. I do have some comments to improve the flow and message of the manuscript.

How many of these sites are being used by the M72 trial?

Further discussion would be useful to discuss the dominance of South Africa in the findings. One sentence is given but I feel this is an important point and some further reasons should be added to the Discussion. Is there any collected information that could shed light on this? What are the implications of this?

6. PLOS authors have the option to publish the peer review history of their article (what does this mean?). If published, this will include your full peer review and any attached files.

Reviewer #1: No

---

## [Author Response · Author response to Decision Letter 0]

13 Mar 2024

15 January 2024

Dear editors of PLOS ONE,

Many thanks for agreeing to consider our appeal and revised manuscript. As indicated by Jianhong Zhou on the 2th December, the editors encouraged re-submission and committed to re-review.

We thank the Reviewers for their comments and suggestions. We have adapted our manuscript according to the Reviewer’s comments and prepared a point-to-point response to their comments.

The present results have not been previously published, the dataset used for this manuscript is publicly available on the EDCTP website https://www.edctp.org/our-work/coordination-tb-vaccine-funded-research/directory-tb-vaccine-clinical-trial-sites-sub-saharan-africa/

We believe that the changes have improved the manuscript and trust that our manuscript now merits publication in the PLOS ONE.

Changes to the current manuscript text compared to the last submission are done using track changes.

Yours sincerely,

Puck Pelzer

https://journals.plos.org/plosone/s/file?id=wjVg/PLOSOne_formatting_sample_main_body.pdf andhttps://journals.plos.org/plosone/s/file?id=ba62/PLOSOne_formatting_sample_title_authors_affiliations.pdf

Answer: The authors have reviewed PLOS ONE’s style requirements and adapted accordingly

“The directory was commissioned and funded by the EDCTP2 programme supported by the European Union and Development of tools to support the coordination of EDCTP TB vaccine funded research.” 

Revised text amended Role of Funder statement: The Funder had no role in study design, data collection and analysis of the data. The Funder was involved in the writing – review and editing – of the manuscript. 

Funded by the European & Developing Countries Clinical Trials Partnership (EDCTP2), supported by the European Union, under the contract “Development of tools and documents to support the coordination of EDCTP TB vaccine funded research.”

Revised text “The directory of TB vaccine clinical trial centres suitable for future TB vaccine studies in Sub-Saharan Africa was conducted by TBVI and KNCV Tuberculosis foundation. The following experts provided advice on essential parameters that were included in the survey: Suzanne Verver, Frank Cobelens, Richard White and Rebecca Harris. We thank Ieva Leimane for her help with survey design, Daniela Pereira for data visualization, Degu Dare, Jeremy Hill, and Max Meis for technical support, and Stephanie Borsboom for project management. Our assessment relied partly on work done by KNCV for Aeras. Last but not least, we wish to thank all contact points from the centres for providing information.”/

“The directory was commissioned and funded by the EDCTP2 programme supported by the European Union and Development of tools to support the coordination of EDCTP TB vaccine funded research.”

4. We noted in your submission details that a portion of your manuscript may have been presented or published elsewhere. [The background data is publicly available on the EDCTP website

https://www.edctp.org/our-work/coordination-tb-vaccine-funded-research/directory-tb-vaccine-clinical-trial-sites-sub-saharan-africa/] Please clarify whether this publication was peer-reviewed and formally published. If this work was previously peer-reviewed and published, in the cover letter please provide the reason that this work does not constitute dual publication and should be included in the current manuscript.

Answer: The present results have not been previously published, the dataset used for this manuscript is publicly available on the EDCTP website https://www.edctp.org/our-work/coordination-tb-vaccine-funded-research/directory-tb-vaccine-clinical-trial-sites-sub-saharan-africa/

Revised text: The dataset used for this manuscript is publicly available on the EDCTP website https://www.edctp.org/our-work/coordination-tb-vaccine-funded-research/directory-tb-vaccine-clinical-trial-sites-sub-saharan-africa/

6. We note that Figures 1 and 4 in your submission contain [map/satellite] images which may be copyrighted. All PLOS content is published under the Creative Commons Attribution License (CC BY 4.0), which means that the manuscript, images, and Supporting Information files will be freely available online, and any third party is permitted to access, download, copy, distribute, and use these materials in any way, even commercially, with proper attribution. For these reasons, we cannot publish previously copyrighted maps or satellite images created using proprietary data, such as Google software (Google Maps, Street View, and Earth). For more information, see our copyright guidelines: http://journals.plos.org/plosone/s/licenses-and-copyright.

1. You may seek permission from the original copyright holder of Figures 1 and 4 to publish the content specifically under the CC BY 4.0 license. 

Answer: The maps were generated on the Visme platform, for which we have a subscription and access to templates. In the Terms and Conditions of Use there are multiple references to copyright, none applies to us. We used the template of the African continent for the shape of the map and used publicly available data for the WHO incidence estimates for input, plus the number of sites surveyed per country. We did not use Google software such as google maps, street view or earth. As there is no copyright owner of these maps we generated, there is no need for a Content Permission Form.

7. We notice that your supplementary 2 table are included in the manuscript file. Please remove them and upload them with the file type 'Supporting Information'. Please ensure that each Supporting Information file has a legend listed in the manuscript after the references list.

Answer: We have added supplementary table 2 to supporting information

Answer: We have reviewed the reference list

Reviewers' comments:

Reviewer's Responses to Questions

Comments to the Author

1. Is the manuscript technically sound, and do the data support the conclusions?

Reviewer #1: Yes

2. Has the statistical analysis been performed appropriately and rigorously?

Reviewer #1: Yes

3. Have the authors made all data underlying the findings in their manuscript fully available?

Reviewer #1: Yes

4. Is the manuscript presented in an intelligible fashion and written in standard English?

Reviewer #1: Yes

5. Review Comments to the Author

Reviewer #1: This is an important manuscript establishing a directory of clinical trial centers across sub-Saharan Africa to assess capacity for conducting late-stage TB vaccine trials. Considering this is the primary location of most future clinical trials this manuscript has important relevance and I recommend publishing. What I find important is the new information provided (available centers) but, perhaps just as valuable, also is locations where centers are not available and capacity is needed. I do have some comments to improve the flow and message of the manuscript.

How many of these sites are being used by the M72 trial?

Answer: The authors are unaware which sites have been selected for the M72 AS01 phase 3 trial.

Further discussion would be useful to discuss the dominance of South Africa in the findings. One sentence is given but I feel this is an important point and some further reasons should be added to the Discussion. Is there any collected information that could shed light on this? What are the implications of this?

Revised text: There is over-representation of South African centres, which could be attributed to the country’s investments in TB vaccine centres. TB vaccine trials need to be conduced in areas with high TB incidence, which are typically in low-resource settings that lack capacity and prior trial experience. The marked prevalence of prior studies in South Africa positions it as an outlier in terms of capacity, elucidating the distinctive nature of its contribution to TB vaccine research. This concentration of research activity in South Africa emphasizes the needs for increased capacity needs for centres across Africa.

---

## [Decision Letter · Decision Letter 1]

21 Jun 2024

PONE-D-23-31310R1The TB vaccine clinical trial centre directory: an inventory of clinical trial centres in sub-Saharan AfricaPLOS ONE

Dear Dr. Pelzer,

Thank you for submitting your manuscript to PLOS ONE. After careful consideration, we feel that it has merit but does not fully meet PLOS ONE’s publication criteria as it currently stands. Therefore, we invite you to submit a revised version of the manuscript that addresses the points raised during the review process.

We look forward to receiving your revised manuscript.

Kind regards,

Felix Bongomin, MB ChB, MSc, MMed, FECMM

Academic Editor

PLOS ONE

Reviewers' comments:

Reviewer's Responses to Questions

**Comments to the Author**

1. If the authors have adequately addressed your comments raised in a previous round of review and you feel that this manuscript is now acceptable for publication, you may indicate that here to bypass the “Comments to the Author” section, enter your conflict of interest statement in the “Confidential to Editor” section, and submit your "Accept" recommendation.

Reviewer #1: All comments have been addressed

Reviewer #2: (No Response)

2. Is the manuscript technically sound, and do the data support the conclusions?

Reviewer #1: Yes

Reviewer #2: Partly

3. Has the statistical analysis been performed appropriately and rigorously? 

Reviewer #1: Yes

Reviewer #2: N/A

4. Have the authors made all data underlying the findings in their manuscript fully available?

Reviewer #1: Yes

Reviewer #2: Yes

5. Is the manuscript presented in an intelligible fashion and written in standard English?

Reviewer #1: Yes

Reviewer #2: No

6. Review Comments to the Author

Reviewer #1: Thank you for selecting me to review this manuscript. The authors have addressed all my concerns. I have no further comments.

Reviewer #2: 1. If the authors have adequately addressed your comments raised in a previous round of review This is my first review for this paper as such i have left this space blank

2. Is the manuscript technically sound, and do the data support the conclusions?  the paper seems to be an attempt to develop a comprehensive directory of clinical trial centers in sub-Saharan Africa (SSA) that can support the coordination of TB vaccine-funded research. The directory includes information on center characteristics, type of disease targets, intervention expertise, and capacity for TB vaccine trials, although the data provided established a foundation to indicate a pool of facilities capable of vaccine research, more could be done to properly highlight the methodological rigor employed to obtaining said directory.

3. Has the statistical analysis been performed appropriately and rigorously?this paper seems to be more of a descriptive exploration of potential tb study sites. Advance statistical analyses were not performed.

4.Have the authors made all data underlying the findings in their manuscript fully available?All dataset are available

5.Is the manuscript presented in an intelligible fashion and written in standard English?  although the manuscript has a generally good flow, more work is needed to make it in line the journal's specifications. More detail is given in reviewer attachment

7. PLOS authors have the option to publish the peer review history of their article (what does this mean?). If published, this will include your full peer review and any attached files.

Reviewer #1: No

Reviewer #2: No

---

## [Author Response · Author response to Decision Letter 1]

5 Sep 2024

Dear Editors of PLOS ONE,

We have submitted both a clean version of the revised manuscript and a version with tracked changes for your review. Additionally, the figures and supplementary materials have been provided as separate files. In the response to the reviewers' comments, we have addressed each point individually, indicating the corresponding changes along with the line numbers where these revisions can be found in the tracked changes version.

We belief this updated manuscript has integrated the feedback by the Reviewers adequately and now merits publication in the PLOS ONE.

Yours sincerely,

Puck Pelzer

---

## [Decision Letter · Decision Letter 2]

24 Sep 2024

The TB vaccine clinical trial centre directory: an inventory of clinical trial centres in sub-Saharan Africa

PONE-D-23-31310R2

Dear Dr. Pelzer,

We’re pleased to inform you that your manuscript has been judged scientifically suitable for publication and will be formally accepted for publication once it meets all outstanding technical requirements.

Kind regards,

Felix Bongomin, MB ChB, MSc, MMed, FECMM

Academic Editor

PLOS ONE

Additional Editor Comments (optional):

Reviewers' comments:

Reviewer's Responses to Questions

**Comments to the Author**

1. If the authors have adequately addressed your comments raised in a previous round of review and you feel that this manuscript is now acceptable for publication, you may indicate that here to bypass the “Comments to the Author” section, enter your conflict of interest statement in the “Confidential to Editor” section, and submit your "Accept" recommendation.

Reviewer #2: All comments have been addressed

2. Is the manuscript technically sound, and do the data support the conclusions?

Reviewer #2: Yes

3. Has the statistical analysis been performed appropriately and rigorously? 

Reviewer #2: N/A

4. Have the authors made all data underlying the findings in their manuscript fully available?

Reviewer #2: Yes

5. Is the manuscript presented in an intelligible fashion and written in standard English?

Reviewer #2: Yes

6. Review Comments to the Author

Reviewer #2: The authors may want to crosscheck basic issues like punctuation and grammar. There are some sentences that end with a comma and a full-stop in the paper.

7. PLOS authors have the option to publish the peer review history of their article (what does this mean?). If published, this will include your full peer review and any attached files.

Reviewer #2: No

---

## [Editor Report · Acceptance letter]

30 Sep 2024

PONE-D-23-31310R2 

PLOS ONE

Dear Dr. Pelzer, 

I'm pleased to inform you that your manuscript has been deemed suitable for publication in PLOS ONE. Congratulations! Your manuscript is now being handed over to our production team.

Kind regards, 

on behalf of

Dr. Felix Bongomin 

Academic Editor

PLOS ONE